# Influence of Health Literacy on Maintenance of Exclusive Breastfeeding at 6 Months Postpartum: A Multicentre Study

**DOI:** 10.3390/ijerph19095411

**Published:** 2022-04-29

**Authors:** María Jesús Valero-Chillerón, Desirée Mena-Tudela, Águeda Cervera-Gasch, Víctor Manuel González-Chordá, Francisco Javier Soriano-Vidal, José Antonio Quesada, Enrique Castro-Sánchez, Rafael Vila-Candel

**Affiliations:** 1Department of Nursing, Universitat Jaume I, Av. de Vicent Sos Baynat, 12071 Castelló, Spain; chillero@uji.es (M.J.V.-C.); cerveraa@uji.es (Á.C.-G.); vchorda@uji.es (V.M.G.-C.); 2Department of Nursing, Universitat de València, 46010 Valencia, Spain; francisco.j.soriano@uv.es (F.J.S.-V.); rafael.vila@uv.es (R.V.-C.); 3Department of Nursing, University of Alicante, 03080 Alicante, Spain; 4Department of Obstetrics and Gynaecology, Hospital Luis Alcanyis, 46819 Xàtiva, Spain; 5Foundation for the Promotion of Health and Biomedical Research in the Valencian Region (FISABIO-SP), 46020 Valencia, Spain; 6Department of Clinical Medicine, Universidad Miguel Hernández, 03202 Elche, Spain; jquesada@umh.es; 7Health Protection Research Unit in Healthcare-Associated Infection and Antimicrobial Resistance at Imperial College London, London W12 0NN, UK; enrique.castro.sanchez@uwl.ac.uk; 8College of Nursing, Midwifery and Healthcare, University of West London, Brentford TW8 9GA, UK; 9Department of Obstetrics and Gynaecology, Hospital Universitario de la Ribera, 46600 Valencia, Spain

**Keywords:** breastfeeding, breastfeeding cessation, early weaning, exclusive breastfeeding, health literacy, nursing, women

## Abstract

Background: International organizations recommend initiating breastfeeding within the first hour of life and maintaining exclusive breastfeeding for the first 6 months. However, worldwide rates of exclusive breastfeeding for 6-month-old infants is far from meeting the goal proposed by the World Health Organization, which is to reach a minimum of 50% of infants. Education is one of the factors affecting the initiation and continuation of breastfeeding, and incidentally, it is also related to lower health literacy. This study explored the influence of health literacy on maintenance of exclusive breastfeeding at 6 months postpartum. Methods: A longitudinal multicenter study with 343 women were recruited between January 2019 and January 2020. The first questionnaire was held during the puerperium (24–48 h) with mothers practicing exclusive breastfeeding, with whom 6-month postpartum breastfeeding follow-up was performed. Socio-demographic, clinical and obstetric variables were collected. Breastfeeding efficiency was assessed using the LATCH breastfeeding assessment tool. The health literacy level was evaluated by the Newest Vital Sign screening tool. A multivariate logistic regression model was used to detect protective factors for early exclusive breastfeeding cessation. Results: One third of the women continued exclusive breastfeeding at 6 months postpartum. Approximately half the participants had a low or inadequate health literacy level. An adequate health literacy level, a high LATCH breastfeeding assessment tool score (>9 points) and being married were the protective factors against exclusive breastfeeding cessation at 6 months postpartum. Conclusion: Health literacy levels are closely related to maintaining exclusive breastfeeding and act as a protective factor against early cessation. A specific instrument is needed to measure the lack of “literacy in breastfeeding”, in order to verify the relationship between health literacy and maintenance of exclusive breastfeeding.

## 1. Introduction

Breastfeeding (BF) offers many health benefits to the mother and the BF infant, both in the short and long term [1]. For example, BF would reduce maternal and infant mortality [2] by 823,000 infants and 20,000 mothers worldwide if exclusive breastfeeding (EBF) up to 6 months was maintained [3]; BF would improve nutritional factors, lower related infant food costs [4,5], and contribute to family and social economic savings by lowering the prevalence of diseases in breastfed newborns (NB) [5,6]. BF also fosters the mother–child bond by encouraging a safe attachment [7] and a better mother–infant relationship [4].

The World Health Organization (WHO) and the United Nations International Children’s Emergency Fund (UNICEF) recommend initiating breastfeeding within the first hour of life and maintaining EBF for the first 6 months. EBF rates at 6 months are low, despite efforts by international organizations [8] to protect and promote this practice [9]. According to the Global Health Observatory data repository [10], only 25% of infants in Europe are breastfed exclusively for the first 6 months [11]. In Spain, EBF prevalence at 6 months is around 16.8%, considerably less than the ~75% EBF rate reported at hospital discharge [12,13].

There are multiple factors for the premature abandonment of breastfeeding. Among these factors, we can find the low weight of the infant, the feeling of lack of milk, smoking, the mother’s lack of knowledge about breastfeeding or the incorporation to work [14,15]. Education is other of the factors affecting the initiation and continuation of breastfeeding (BF) [9,16,17], and incidentally it is also related to health literacy (HL) [18,19]. The concept of HL emerged in the 1970s and has been continuously refined since then [20,21]. Health literacy is currently defined as “an individual’s ability to obtain and translate knowledge and information in order to maintain and improve health in ways that are appropriate to the individual and community context” [22]. A low HL level has been linked to difficulties understanding healthcare information and to poor therapeutic concordance, which in turn increases costs and leads to an inefficient use of healthcare resources [22,23].

Likewise, women’s HL levels can also have an effect on their children’s health during pregnancy and after birth [24,25]. As for the decision to breastfeed, the percentage of mothers who decide to BF rises with their HL level [26]. In one small study, health literacy was found to be a protective factor for breastfeeding [27]. Consistent with these results, the aim of this study was to explore the influence between the level of health literacy and the maintenance of exclusive breastfeeding at six months postpartum.

## 2. Materials and Methods

### 2.1. Design and Setting

A longitudinal multicenter study was carried out at three hospitals in the Valencian Community (Spain): The General University Hospital of Castellón (Department of Health, Castellón); the University La Ribera Hospital (Department of Health, La Ribera); the Lluís Alcanyís Hospital of Xátiva (Department of Health, Xátiva-Ontinyent). These hospitals were either reference centers for their province (General University Hospital of Castellón and University La Ribera Hospital) or were in a rural area with large catchment populations (Lluís Alcanyís Hospital). Overall, the participating hospitals served 600,000 people.

### 2.2. Sample

The target population comprised women registered with the Departments of Health of Castellón, La Ribera and Xátiva-Ontinyent, whose birth was at one of the participating hospitals, and who had opted for EBF on discharge.

Systematic sampling of women admitted to hospital during clinical puerperium was conducted by randomly selecting one in every three puerperal women on the maternity ward every Monday. All women who wished to participate in the study were recruited, and they were asked to sign informed written consent. Mothers who were older than 18 years and had no health problems associated and/or puerperal complications at discharge were included in the study.

Some situations may make it difficult to initiate lactation. For this reason, twin pregnancies, and multiple and/or premature pregnancies, and/or congenital anomalies detected in the first 24 h, and/or newborns admitted in neonatal intensive care unit were excluded [16,17]. We excluded women with cognitive impairments, language barriers, or illiteracy (not able to read). Illiterate women were excluded from the study as they would be unable to complete the self-administered health literacy screening tools [25]. Finally, we also excluded mothers who we were unreachable by telephone after three attempts at 6 months postpartum.

We assumed that if the proportion of women with limited HL at baseline were 45%, the EBF cessation rate in the adequate HL group was 40% and, to detect a difference between groups of 15% on EBF cessation, as well as a 0.05 confidence level and 80% statistical power, 350 women were required [27]. Considering a 10% attrition rate, the final sample size was estimated at 385 women. The sample size calculation was performed by EPIDAT v.3.1, Santiago de Compostela, Spain.

### 2.3. Data Collection, Main Variables and HL Measure

The participating hospitals attend an average of 1600 births per year in Castellón, 1400 births in La Ribera, and 700 births in Xàtiva-Ontinyent. Therefore, the number of participants in each department has been influenced by the number of births attended in each hospital.

Printed questionnaires were used to collect data. Participants were recruited between January 2019 and January 2020 during clinical puerperium (24–48 h after giving birth). One researcher per participating center oversaw the first data collection, except for the HL screening tool, which women self-administered before discharge from hospital. The BF follow-up at 1, 2 and 4 postpartum months was performed by the same researcher by consulting each participants’ electronic health records. Finally, when breastfed infants were 6 months old, mothers were telephoned to document their feeding type. 

BF efficacy was evaluated using the LATCH breastfeeding assessment tool. This questionnaire has been validated in Spanish [28] and contains five items (“Latch”, “Audible swallowing”, “Type of nipple”, “Comfort” and “Hold–positioning”). Each item is scored numerically (0–2), where 0 is the worst possible and 2 the best possible situation. A score of 8–10 reflects effective breastfeeding. During fieldwork, BF efficacy was evaluated with this instrument by the researcher in charge at each participating hospital before hospital discharge. 

While contacting mothers, they were asked whether they continued EBF. If their answer was negative, they were asked about the feeding type they provided and how long they had practiced EBF. The questions were: 1. Are you still exclusively breastfeeding your baby? 2. If not, for how long did you exclusively BF your baby? Finally, feeding type information and duration were recorded in their electronic medical records. The researchers attempted a maximum of three calls per participant and followed a pre-established script to reduce data loss as much as possible and maximize data quality. Feeding type was classified as [16,29]: 1. EBF means that infant receives only breast milk or expressed milk; 2. Formula milk; 3. Mixed BF (combination of breast milk and formula milk). BF status was recorded at hospital discharge (48–72 h), and at 1, 2, 4 and 6 months after giving birth. Early EBF cessation was considered if it occurred before 6 months postpartum (yes/no), as set out by the World Health Organization among its 2025 targets [30]. 

The HL was explored through an interview at discharge and was measured by the Newest Vital Sign (NVS) questionnaire validated in Spanish, with acceptable internal consistency (α = 0.69) [31]. This self-administered questionnaire contains six questions about a nutritional ice cream label. One point is scored per correct answer [32]. Questions are freely answered and do not lead participants to any expected response type. It classifies the HL level according to the overall score as “adequate” (4–6 points) or “limited” (<4 points).

### 2.4. Data Analysis

A descriptive analysis was performed using absolute and relative frequencies for qualitative variables (socio-demographic and obstetric variables), and the mean and standard deviation (±SD) for quantitative variables. The HL-related factors and those associated with EBF cessation at 6 months were analyzed using 2 × 2 tables, the chi-squared test (χ^2^) for qualitative variables and by comparing the means for quantitative variables via the one-factor analysis of variance (ANOVA) or the Student’s *t*-test, respectively.

The magnitudes of the associations with EBF cessation at 6 months were dealt with by the fit of the multivariate logistic models. The odds ratio (OR) was estimated along with their 95% confidence intervals (95% CIs). A stepwise procedure based on AIC’s criterion (Akaike Information Criterion) was followed to select variables. Data analysis was performed on SPSS v.25.0 statistical package (IBM Corp. Released 2018. IBM SPSS Statistics for Windows, Armonk, NY, USA) and R (R project 2019, Version 3.5.1, Vienna, Austria). As the analysis included two variables, NVS and EBF cessation, the level of significance was adjusted by the Bonferroni method to *p* < 0.025.

## 3. Results

Of 391 participants initially recruited, 48 (12.3%) were later excluded: 42 (87.5%) due to follow-up loss and six (12.5%) because they did not wish to continue in the study during follow-up. The homogeneity between those who were included and those who were excluded or lost from participation was analyzed. There were no significant differences in age, age at first pregnancy, gestational age at delivery, health literacy level by NVS or country of origin between the group that was included and the group that was excluded in this study. 

The final sample size was 343 women who reported EBF when discharged from hospital, and who were included in the BF follow-up until breastfed infants were 6 months old. 

### 3.1. Socio-Demographic Characteristics

The participants’ mean age was 32.5 years (±5.3). The mean gestational age at birth was 39 + 3 weeks (±1.1), and the mean birth weight was 3301.2 g (±464.5). Table 1 shows the other socio-demographic variables included in this study. 

### 3.2. BF-Related Variables

The mean LATCH breastfeeding assessment tool score for BF efficiency was 8.8 out of 10 points (±0.9). The 6-month EBF rate was 34.1% (117/343), with 65.9% (226/343) for EBF cessation before 6 months (Figure 1). 

### 3.3. HL Level

Of all participants, 47.8% (164/343) had a limited HL level. The factors associated with a limited HL level were, a lower level of education (*p* < 0.001), being unemployed or not looking for a job (*p* = 0.003), and not born in Spain (*p* < 0.001). However, the mother’s older age (*p* < 0.001) was associated with a higher HL level (Table 2). Figure 2 indicates the distribution of HL levels in relation to EBF at 6 months. 

Table 3 shows the relationship between the collected variables and their association with EBF cessation at 6 months. The variables associated with early EBF cessation were a limited HL level (*p* < 0.001), being a single, separated, divorced mother (*p* < 0.001), having a lower level of education (*p* = 0.022), and obtaining a lower LATCH breastfeeding assessment tool score (*p* < 0.001). Conversely, a mean score of 9.19 (±0.85) for BF efficiency at hospital discharge, as measured by the LATCH breastfeeding assessment tool, presented a statistically significant association (*p* < 0.001) with continuing with EBF until infants were 6 months old. 

### 3.4. Variables Related to Early EBF Cessation 

The multivariate regression model shown in Table 4 for EBF cessation before 6 months suggests that a limited HL level is associated with more than twice the probability of EBF cessation before 6 months compared to an adequate HL level adjusted by mother’s age, level education, civil status and the LATCH breastfeeding assessment tool. Both being married and obtaining a higher LATCH breastfeeding assessment tool score were also protective factors against EBF cessation before infants were 6 months old.

## 4. Discussion

The present study focuses on continuation of EBF until infants are 6 months old and explores influential factors, namely HL levels.

One of the WHO’s goals for 2025 is to reach EBF rates of at least 50% until infants are 6 months old [30]. Worldwide EBF rates at 6 months fall short of this recommendation [33]. Between 2006 and 2012 in Europe, it was estimated that only 25% of breastfed infants received EBF for the first 6 months of life [11]. According to the European Health Information Gateway [34], EBF rates at 6 months were 58.3% in Italy (2011), 53.9% in Portugal (2013) and 58.4% in Spain (2017). However, more recent studies carried out in Spain report considerably lower EBF infants until the age of 6 months, ranging from 16.8% [12], 21.6% [35], or 31.4% [36], to 43% [37]. 

Different studies have reported an association between mothers’ level of education and continuing with EBF and showing that the higher the level of education, the longer that EBF lasts [35,36,37], in line with our results. Other authors have established an association between level of education and HL levels [18,19]. Although it may seem that a low educational level could be associated with a low HL, this relationship does not always have to be observed [38]. A relation was also recently found between HL levels and continuing EBF in a pilot study; however, the follow-up period only covered 4 months [25]. Therefore, the present study verifies a statistically significant association between limited HL level and EBF cessation before 6 months in line with previous studies [27], and observed that the probability of EBF cessation was more than two-fold compared to the mothers with an adequate HL level. 

Previous studies have related found an association with mothers’ age and early EBF cessation [9,35,37]. In agreement with results hitherto reported [39], we noted a statistically significant association between being older and EBF rates at 6 months postpartum. This association might be due to ongoing family support, better socio-economic status or a higher level of knowledge about BF benefits, as other research has shown [40,41,42,43]. There are also reports indicating that those families with single, separated, or divorced mothers, the probability of EBF cessation before 6 months postpartum more than doubles. For continuing EBF, several studies have verified that family support [44] and having a partner are key factors [45,46]. Other authors have reported how the probability of EBF cessation before 6 months postpartum more than doubles in families with single, separated, or divorced mothers [47,48]. Women’s immediate environment (family, friends and neighbours) is the most influential social support network in shaping pregnant women’s expectations and decisions about pregnancy, labour and nursing [49]. However, the NVS tool does not incorporate those social aspects, unlike other tools such as the Health Literacy Questionnaire [50], so their influence on the HL of breastfeeding women remains to be clarified [51]. It is noteworthy that being older with a first pregnancy also showed a statistically significant association with an adequate HL level. It was not surprising that the two variables contributing to continuing EBF, namely an adequate HL level and older maternal age, were also closely interrelated, as seen in a recent study in Spain [43]. Nevertheless, future studies are needed to corroborate the relationship between being older with first pregnancy and continuing EBF and a higher HL level, and the factors that could influence the relationship between both these variables must also be explored.

The average LATCH score was high with a small standard deviation, which suggests that the majority of the study population was breastfeeding effectively or nearly effectively. It is worth stressing the predictive capacity of the LATCH breastfeeding assessment tool. Different studies have measured BF efficacy both postpartum and before hospital discharge. These studies showed that BF efficacy can be effectively evaluated using LATCH [52], and its predictive performance is high at 6 weeks postpartum [53,54,55]. The present study revealed that high LATCH breastfeeding assessment tool scores were significantly associated with a lower probability—almost half—of EBF cessation before breastfed infants were 6 months. As the LATCH breastfeeding assessment tool seems to be useful, future studies should take advantage of these findings to relate the LATCH scores with continuation of EBF in the longer term. However, the LATCH could be further refined to incorporate elements such as mother/infant interaction [56]. This tool has major flaws, including the inability of the user to assign different scores per breast (e.g., if one nipple is flat and the other is everted), the lack of representation of infant’s oral anatomy and functionality.

Moreover, women often seek support beyond their home if it is not available there. However, more studies need to be conducted to corroborate the association between family support and continuing EBF to 6 months.

Despite the need for more robust studies to determine the association between level of health literacy and maintenance of exclusive breastfeeding at 6 months postpartum, this study shows a profile of women that should not go unnoticed by health professionals caring for women during the perinatal period. According to the results of this study, the profile of women who are more likely to early EBF cessation would be those who are younger, single, separated or divorced, who have a limited level of LH as measured by the NVS instrument, as well as those who have a LATCH score below 9. Different interventions in low HL women have been designed to improve the outcomes and experiences in relation to breastfeeding promotion. Interventions based on education alone are inadequate to improve low HL, and multidimensional and multidisciplinary methodologies are needed to identify the best strategy. In addition, more research is needed in order to improve this knowledge, due to the low quality of the evidence of the studies [57]. Therefore, the development of interventions to improve LH in relation to BF seems to be an interesting future line of research.

### Limitations

One limitation of this study is the information bias related to the reasons for BF cessation. This information could not be collected because the reason for breastfeeding cessation is not always reflected on electronic health records. Several studies have widely reported on reasons explaining why BF cessation is poorly registered [13,58]. Although this information is relevant, it is beyond the scope of our study. The three health departments use the same electronic medical record that is also common to the Valencian community (ABUCASIS II), so the information bias was controlled because the quality of the records was uniform between the three hospitals. In addition, marital status was taken into account, but other consensual unions were disregarded. Future research should take this into account.

The percentage of the women excluded by criteria or lost were similar in the three health departments. We obtained a heterogeneous sample between the three health departments, explained by the number of annual deliveries, and the availability of the researcher responsible for the hospital to recruit the sample, being lower in Xàtiva health department than the others. Although the analysed sample could be larger than that obtained, the decision that a single researcher for each centre would carry out the evaluations in the clinical puerperium, and later take charge of the follow-up, analysing the medical records and making the telephone calls at six months, we believe that the information bias is reduced, avoiding recording and interpretation errors. 

The strengths of this study include the characteristics of the sample, as it was randomised and representative. 

This study identified that a limited HL level was a statistically significant risk factor for EBF cessation before breastfed infants were 6 months old. It should be noted that NVS is designed to measure HL in the general population; perhaps the use of a specific tool designed for perinatal women would provide different results. The screening tool herein employed was the Newest Vital Sign (NVS), whose validated Spanish version presents acceptable internal consistency (Cronbach’s α = 0.69) [31]. This instrument has been previously employed to assess the association between HL levels and continuing EBF until breastfed infants were 4 months old [25]. Nonetheless, it is worth indicating that continuing BF might be more closely related to HL while breastfeeding compared to general HL level. 

## 5. Conclusions

The present study demonstrates how adequate HL levels can influence the maintenance of EBF, acting as a protective factor against early cessation. Therefore, one recommendation is to include HL level as a relevant risk factor when adopting preventive strategies to increase the EBF rate at 6 months postpartum in order to move towards international recommendations. Further exploration of the components of HL and its relationship to factors influencing cessation of EBF will be important. 

## Figures and Tables

**Figure 1 ijerph-19-05411-f001:**
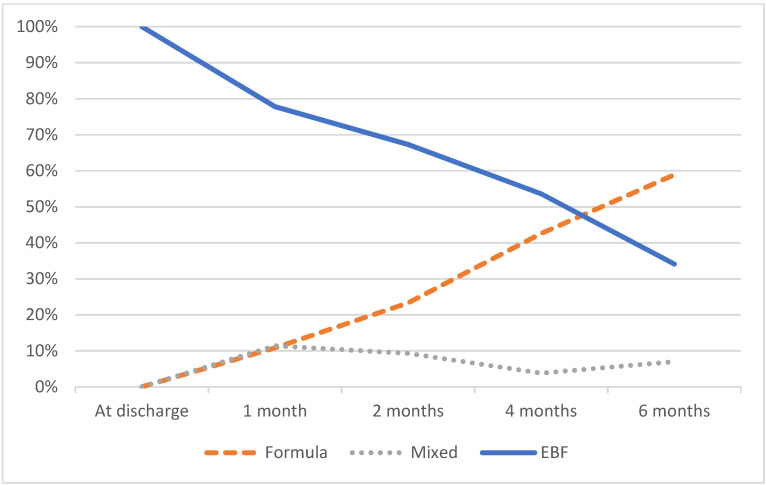
Feeding type during the study period.

**Figure 2 ijerph-19-05411-f002:**
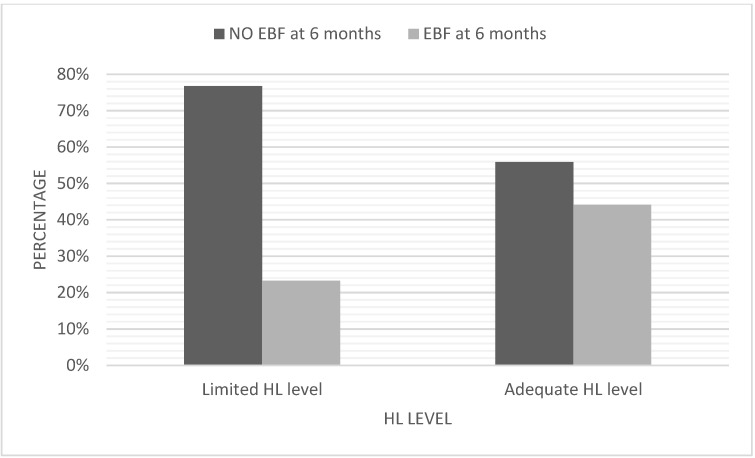
Distribution of HL levels in relation to EBF at 6 months (N = 343).

**Table 1 ijerph-19-05411-t001:** Characteristics of the included females (n = 343).

		n	%
EBF 6 months	Yes	117	34.1%
	No	226	65.9%
NVS	Adequate HL level	179	52.2%
	Limited HL level	164	47.8%
Department of Health	La Ribera	216	63.0%
	Xátiva-Ontinyent	24	7.0%
	Castellón	103	30.0%
Civil status	Married	216	63.0%
	Single, separated, divorced	127	37.0%
Level of education	Primary or lower	97	28.3%
	1st cycle, Secondary	58	16.9%
	2nd cycle, Secondary	86	25.1%
	University diploma	40	11.7%
	Graduate	62	18.1%
Pregnant women’s occupation	Businesswoman/Professional	35	10.2%
	Employee	203	59.2%
	Unemployed	84	24.5%
	Not looking for a job	21	6.1%
Country of origin	Spain	278	81.0%
	Foreign	65	19.0%
Partner’s occupation	Employee	273	79.6%
	Businessperson/Professional	31	9.0%
	Others	39	11.4%
Parity	One	176	51.3%
	Two or more	167	48.7%
Skin-to-skin contact at birth	No	38	11.1%
	Yes	305	88.9%
Birth type	Spontaneous	202	58.9%
	Instrumented	62	18.1%
	STC	79	23.0%
Risk pregnancy	Low risk	236	68.8%
	High risk *	107	31.2%
		n	Mean (SD)
Mother’s age	(years)	343	32.5 (5.2)
Age with first pregnancy	(years)	343	29.8 (5.7)
Gestational week at birth	(weeks)	343	39.3 (1.1)
LATCH score	(0 to 10)	343	8.8 (0.9)
Birth weight	(grams)	343	3301.2 (464.5)

EBF: exclusive breastfeeding; NVS: Newest Vital Sign; STC: segment transverse caesarean; LATCH: Latch audible type comfort hold. * High risk pregnancy = Preeclampsia, Gestational diabetes, Obesity, Low body mass index, Mother age > 35 years, Assisted Reproductive Treatment, Thyroid pathology, Small for gestational age, large for gestational age, fetal growth restriction, Autoimmune diseases, Previous cesarean section, Previous preterm birth and Hepatitis Virus infection.

**Table 2 ijerph-19-05411-t002:** Relation between HL levels (NVS) and the studied variables.

		Adequate HL Level		Limited HL Level		
		n	%	n	%	*p*-Value ^1^
Department of Health	La Ribera	107	49.5	109	50.5	0.424
	Xátiva-Ontinyent	13	54.2	11	45.8	
	Castellón	59	57.3	44	42.7	
Civil status	Married	114	52.8	102	47.2	0.775
	Single, separated, divorced	65	51.2	62	48.8	
Level of education	Primary or lower	35	36.1	62	63.9	<0.001
	1st cycle, Secondary	19	32.8	39	67.2	
	2nd cycle, Secondary	44	51.2	42	48.8	
	University diploma	32	80.0	8	20.0	
	Graduate	49	79.0	13	21.0	
Pregnant women’s occupation	Businesswoman	23	65.7	12	34.3	0.003
	Employee	113	55.7	90	44.3	
	Unemployed	39	46.4	45	53.6	
	Not looking for a job	4	19.0	17	81.0	
Country of origin	Spain	164	59.0	114	41.0	<0.001
	Foreign	15	23.1	50	76.9	
Partner’s occupation	Employee	148	54.2	125	45.8	0.183
	Businessperson	16	51.6	15	48.4	
	Others	15	38.5	24	61.5	
Parity	One	99	56.3	77	43.8	0.122
	Two or more	80	47.9	87	52.1	
Skin-to-skin contact at birth	No	19	50.0	19	50.0	0.775
	Yes	160	52.5	145	47.5	
Birth type	Spontaneous	95	47.0	107	53.0	0.068
	Instrumented	38	61.3	24	38.7	
	STC	46	58.2	33	41.8	
Risk pregnancy	Low risk	115	48.7	121	51.3	0.057
	High risk	64	59.8	43	40.2	
		n	Mean (SD)	n	Mean (SD)	*p*-value ^2^
Mother’s age	(years)	179	33.5 (4.8)	164	31.4 (5.5)	0.001
Gestational week at birth	(weeks)	179	39.3 (1.1)	164	39.4 (1.1)	0.765
LATCH score	(0 to 10)	179	8.9 (0.9)	164	8.7 (0.9)	0.037
Birth weight	(grams)	179	3281 (476.6)	164	3311 (452.2)	0.684

EBF: exclusive breastfeeding; NVS: Newest Vital Sign; STC: segment transverse caesarean; LATCH: Latch audible type comfort hold: ^1^ Chi-square test; ^2^ Student’s *t*-test.

**Table 3 ijerph-19-05411-t003:** Factors related to EBF cessation before 6 months.

		EBF 6 Months: Yes		EBF 6 Months: No		
		n	%	n	%	*p*-Value ^1^
NVS	Adequate HL level	79	44.1	100	55.9	<0.001
	Limited HL level	38	23.2	126	76.8	
Department of Health	La Ribera	65	30.1	151	69.9	0.105
	Xátiva-Ontinyent	11	45.8	13	54.2	
	Castellón	41	39.8	62	60.2	
Civil status	Married	89	41.2	127	58.8	<0.001
	Single, separated, divorced	28	22.0	99	78.0	
Level of education	Primary or lower	25	25.8	72	74.2	0.022
	1st cycle, Secondary	22	37.9	36	62.1	
	2nd cycle, Secondary	29	33.7	57	66.3	
	University diploma	22	55.0	18	45.0	
	Graduate	19	30.6	43	69.4	
Pregnant women’s occupation	Businesswoman	11	31.4	24	68.6	0.850
	Employee	73	36.0	130	64.0	
	Unemployed	26	31.0	58	69.0	
	Not looking for a job	7	33.3	14	66.7	
Country of origin	Spain	100	36.0	178	64.0	0.133
	Foreign	17	26.2	48	73.8	
Partner’s occupation	Employee	95	34.8	178	65.2	0.330
	Businessperson	7	22.6	24	77.4	
	Others	15	38.5	24	61.5	
Parity	One	58	33.0	118	67.0	0.643
	Two or more	59	35.3	108	64.7	
Skin-to-skin contact at birth	No	9	23.7	29	76.3	0.151
	Yes	108	35.4	197	64.6	
Birth type	Spontaneous	62	30.7	140	69.3	0.255
	Instrumented	23	37.1	39	62.9	
	STC	32	40.5	47	59.5	
Risk pregnancy	Low risk	76	32.2	160	67.8	0.268
	High risk	41	38.3	66	61.7	
		n	Mean (SD)	n	Mean (SD)	*p*-value ^2^
Mother’s age	(years)	117	33.2 (4.5)	226	32.1 (5.6)	0.049
Gestational week at birth	(weeks)	117	39.3 (1.0)	226	39.4 (1.1)	0.498
LATCH score	(0 to 10)	117	9.2 (0.8)	226	8.7 (0.9)	<0.001
Birth weight	(grams)	117	3328 (474.7)	226	3286 (459.6)	0.637

EBF: exclusive breastfeeding; NVS: Newest Vital Sign; STC: segment transverse caesarean; LATCH: Latch audible type comfort hold: ^1^ Chi-square test; ^2^ Mann–Whitney test.

**Table 4 ijerph-19-05411-t004:** Multivariate logistic model for EBF cessation before 6 months.

		OR	95% CI	*p*-Value
NVS	Adequate HL level	1		
	Limited HL level	2.52	(1.45–4.36)	0.001
Civil status	Married	1		
	Single, separated, divorced, widowed	2.32	(1.34–4.01)	0.003
Level of education	Primary or lower	1		
	1st cycle, Secondary	0.62	(0.30–1.31)	0.210
	2nd cycle, Secondary	0.86	(0.43–1.73)	0.664
	University diploma	0.51	(0.22–1.20)	0.124
	Graduate	1.11	(0.50–2.50)	0.799
Mother’s age	(years)	0.99	(0.94–1.04)	0.569
LATCH score	(range 6 to 10)	0.53	(0.40–0.71)	<0.001

n model = 343; n EBF cessation = 226; ROC area = 0.7401, 95% CI: 0.6868–0.7933; Likelihood Ratio Test = 58.0 (*p* < 0.001). NVS: Newest Vital Sign; OR: Odds Ratio; 95% CI: 95% confidence interval.

## Data Availability

Please note that the database is in an open repository. You can access the data through this link: http://hdl.handle.net/10234/196606 (accessed on 20 February 2022).

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
