# Peer review of "Influence of Health Literacy on Maintenance of Exclusive Breastfeeding at 6 Months Postpartum: A Multicentre Study"

_ijerph, 2022, doi:10.3390/ijerph19095411_

Round 1

Reviewer 1 Report

The article presents a very pertinent, socially relevant and impactful topic for public health and social welfare. The quality of the study is reinforced by its longitudinal perspective.
The structure of the article, the size of the different sections and their contents are presented in a balanced and sufficiently detailed way.
The methodology is well presented, with relevant information. The sample is robust and presents good potential for analysis.
The authors present, in a very honest way, the limitations of the study, which reinforces the quality of the work.
Nevertheless, I believe that the article could be improved in some aspects, namely:
Introduction
- The framework could be strengthened, regarding the influence of sociodemographic characteristics of mothers on breastfeeding outcomes.
- They present the definition of 'health literacy' (in the introduction) and, further on, explain its operationalisation. But they do not discuss this operationalisation (could it be reductive?) and the possibility of crossing these results with women's education, for example. 
Materials and Methods
- The choice of the regions selected for the study is not justified, and those regions are not characterised.
- The exclusion of women with certain characteristics is pointed out. Won't some exclusions (such as illetarate women) condition the very results of the study?  

Results
- In the sample characteristics, did they not consider marital status in addition to marital status? How were the cases of consensual union considered?
- In occupation, did they mix the categories of employed/unemployed/inactive, with an occupational catergory? What about the remaining ones? And who answered employed, could not select this occupation? And in the case of the partner, were the categories other?

Author Response

Dear reviewer,

We would like to thank you for your valuable contributions, which have enabled us to substantially improve the quality of the manuscript.

Please find attached a document reflecting each of the changes made for each of the contributions.

We hope that the improvements made meet your expectations as a reviewer.

Yours sincerely.

Reviewer 2 Report

It was a pleasure for me to review this manuscript that studies the influence of health literacy for the maintenance of breastfeeding.

First of all, I would like to congratulate the authors for having chosen such an interesting topic because exclusive breastfeeding as a way of feeding the newborn is increasingly difficult for many mothers to maintain over time.

In general, the manuscript seems quite good to me, both at the content level and at the methodological level.

The introductory section seemed correct to me. It immediately places the reader in the context of the problem and exposes the scientific evidence published to date.

The material and methods section perfectly describes the sample, its shape and size chosen to be a representative sample. In addition, the measurement instruments used and how the data were collected were clearly explained.

At the end of the material and methods section, information related to ethical aspects must be included. Please add this information.

The methodology seemed correct to me. For the bivariate study, parametric and non-parametric tests were used, depending on the case. Perhaps it would have been convenient to indicate which variables followed a normal distribution and which did not.

Undoubtedly, the strongest point of this research was the multivariate logistic study, establishing the factors that influenced the maintenance of breastfeeding.

The discussion section also seemed correct, comparing the results obtained with similar studies and establishing the limitations found in the study.
The conclusions section, although somewhat brief, supports the results and opens lines of research on this topic for the future.

Thank you very much.

Kind regards

Author Response

Dear Reviewer,

We would like to thank you for your feedback on the manuscript.

We have considered your input and made the appropriate changes, as reflected in the attached document.

We hope you will be pleased with the result.

Yours sincerely.

Reviewer 3 Report

An interesting study. Requires some editing for English language and readability. Would like to see some elements in the methods clarified and some more discussion on ways to further elucidate the reasons for cessation in EBF - or more suggestions for ways to either increase HL or targeted recommendations to reduce the risk of cessation for those at higher risk. See attached PDF with highlighted areas and comments.

Author Response

(The authors gave the same response as above.)
